# From Snake Venom’s Disintegrins and C-Type Lectins to Anti-Platelet Drugs

**DOI:** 10.3390/toxins11050303

**Published:** 2019-05-27

**Authors:** Philip Lazarovici, Cezary Marcinkiewicz, Peter I. Lelkes

**Affiliations:** 1School of Pharmacy Institute for Drug Research, Faculty of Medicine, The Hebrew University of Jerusalem, Jerusalem 91120, Israel; 2Department of Bioengineering, College of Engineering, Temple University, Philadelphia, PA 19122, USA; cmarcink@temple.edu (C.M.); pilelkes@temple.edu (P.I.L.)

**Keywords:** snake venom, Tirofiban, Eptifibatide, Vipegitide, anti-platelet drug, acute coronary syndrome, percutaneous coronary intervention, clinical trial, adverse effect

## Abstract

Snake venoms are attractive natural sources for drug discovery and development, with a number of substances either in clinical use or in research and development. These drugs were developed based on RGD-containing snake venom disintegrins, which efficiently antagonize fibrinogen activation of αIIbβ3 integrin (glycoprotein GP IIb/IIIa). Typical examples of anti-platelet drugs found in clinics are Integrilin (Eptifibatide), a heptapeptide derived from Barbourin, a protein found in the venom of the American Southeastern pygmy rattlesnake and Aggrastat (Tirofiban), a small molecule based on the structure of Echistatin, and a protein found in the venom of the saw-scaled viper. Using a similar drug discovery approach, linear and cyclic peptides containing the sequence K(R)TS derived from VP12, a C-type lectin protein found in the venom of Israeli viper venom, were used as a template to synthesize Vipegitide, a novel peptidomimetic antagonist of α2β1 integrin, with anti-platelet activity. This review focus on drug discovery of these anti-platelet agents, their indications for clinical use in acute coronary syndromes and percutaneous coronary intervention based on several clinical trials, as well as their adverse effects.

## 1. Introduction

### 1.1. Integrins

Integrins are a large group of receptors composed of different combinations of the α and β chains. These receptors mediate cell adhesion to extracellular matrix (ECM) proteins and are involved in cell-cell interactions. ECM protein ligands bind integrins through the Arg-Gly-Asp (RGD) sequence motif, surrounded by disulphide bonds and neighbored by charged amino acids. This motif is present in ECM proteins such as vitronectin, fibronectin, osteopontin, and fibrinogen. Sequences neighboring the RGD motif are important for integrin recognition specificity. Integrin receptors activation upon binding the respective ECM ligand promotes intracellular signaling. As a consequence, they have diverse roles in cell migration and wound healing, cell differentiation, and apoptosis and can also regulate the metastatic and invasiveness of tumor cells. The regulation of integrin activity is complex. They are activated by extracellular ligands that initiate intracellular signals which convert (‘inside-out’) the extracellular domains to a high-affinity phase by a conformational change. Binding of ECM ligands to the integrin receptor propagates signaling across the membrane (‘outside-in’) to activate cytoplasmic protein kinases and cytoskeletal-signaling cascades. These biochemical activities control fundamental cell physiology processes such as survival adherence, movement, growth, and differentiation [1,2,3]. In addition, integrins also interact with growth factors receptors to regulate cell migration, blood vessel development, and angiogenesis. Importantly, the understanding of the mechanism of action of integrins paralleled the discovery of proteins from snake venoms, known as the disintegrins, which function as potent inhibitors of platelet aggregation and integrin receptor-dependent cell adhesion [4,5]. 

### 1.2. Disintegrins

Disintegrins are a group of low molecular weight, polypeptides folded by cysteine bridges that modulate cell adhesion, migration, apoptosis, platelet aggregation, and angiogenesis. They are naturally derived by proteolytic processing from metalloproteinase precursors and carrying the integrins’ recognition motifs RGD, KGD, WGD, VGD, MGD, RTS, KTS [4,5,6]. Currently, disintegrins are divided into several groups. The major group is represented by short disintegrins which contain 41–51 residues and folded by four cysteine bridges (e.g., echistatin-RGD; obtustatin-KTS); the second group consist of members with higher molecular weight containing approximately 70 amino acids and folded by six cysteine bridges (e.g., barbourin-KGD, flavoviridin-RGD and atrolysin E-MVD); the third group includes disintegrins with approximately 84-residue and cross-linked by seven cysteine bridges (e.g., bitistatin-RGD). The fourth group represents macromolecular complexes formed by identical (homodimers) or different (heterodimers) protein monomers, which are usually non-covalently bound composed of approximately 67 amino acids. They contain ten cysteines (e.g., EC3A-VGD) involved in the formation of four intra-chain and two inter-chain cysteine bridges. It is worth paying attention to the fact that the disintegrin function and specificity depends on the appropriate cysteines bridges required for folding and conformational exposure of the three amino acids binding motif responsible for the inhibition of platelet aggregation or endothelial cell function. Accordingly, disintegrins have been used as important biochemical tools in pharmacological research, in the development of anti-platelet and anti-angiogenesis drugs [7,8]. Most of the monomeric disintegrins contain RGD, KGD, MVD, MGD, or WGD and display antagonistic activity towards integrins such as αIIbβ3 (fibrinogen receptor), αvβ3 (vitronectin receptor) and α5β1 (fibronectin receptor). Some RGD disintegrins inhibit α3β1, α6β1, and α7β1 laminin receptors [9] affecting neutrophils migration, tumors, and skeletal muscle functions. Another group containing the MLD motif interacts with lymphocytes α4β1, α4β7, and α9β1 integrins [10]. Finally, KTS/ RTS disintegrins are selective inhibitors of α1β1 collagen receptor [11]. 

### 1.3. Platelets’ Integrins 

Platelets express several β1 and β3 containing integrins: α2β1 (collagen receptor), α5β1 (fibronectin receptor), α6β1 (laminin receptor), αIIbβ3 (fibrinogen receptor, also known as GP IIb/IIIa), and αVβ3 (vitronectin receptor). The dominant integrin on the platelet membrane is αIIbβ3, with a supplementary cytoplasmic pool that is exposed on the platelet surface upon activation. It binds RGD containing ECM ligands such as fibrinogen, fibrin, von Willebrand factor (vWF), fibronectin, thrombospondin, and vitronectin [12]. αIIbβ3 exists in several interconvertible states: an inactive (resting) state that does not bind ECM protein ligands and two active ligand-binding states that differ in their affinity for fibrinogen and in the mechanical stability of fibrinogen [13]. Upon activation, integrin αIIbβ3 mediates platelet adhesion and aggregation on the exposed ECM protein of the injured vessel wall by promoting crosslinking between neighboring platelets through its main ligand fibrinogen, or at high shear stress through vWF, resulting with thrombus formation. In unstimulated platelets, the integrin is in a “low-affinity” state characterized by a bent conformation with unexposed RGD. Upon platelet stimulation, “inside-out” signaling causes a conformational switch where integrins undergo reorganization into a “high-affinity” state exposing the binding site for RGD. In some cardiovascular pathology, platelet activation and aggregation cause thrombotic vessel occlusion and subsequent tissue damage, as in myocardial infarction (MI) and brain stroke. Due to its importance in platelet aggregation, inhibiting RGD-dependent integrins such as αIIbβ3 has become an appealing target for therapy of ischemic cardiovascular events [14,15,16]. Therefore, is not surprising that active three amino acid motifs in the structure of disintegrins have been used as a lead structure to design compounds that bind and block integrin receptor αIIbβ3. Figure 1 illustrates the source of snake venom proteins used as lead compounds for the development of drugs acting as antagonists of platelets integrins and therefore pharmacologically defined as anti-platelet aggregation/antithrombotic drugs. Integrilin is a synthetic cyclic peptide adjusted from the snake venom disintegrin named Barbourin. Aggrastat is a non-peptide, synthetic compound adjusted from Echistatin. Both compounds are pharmacological antagonists of αIIbβ3 integrin, competing with the binding of fibrinogen and vWF. Another effective blocker of αIIbβ3 is the neutralizing monoclonal antibody Abciximab. These three drugs were approved for therapy of acute coronary syndromes and percutaneous coronary interventions emphasizing the utility of targeting αIIbβ3 selective disintegrins in anti-platelet therapy [17,18].

### 1.4. Inhibition of Platelet Aggregation in Acute Coronary Syndromes and Percutaneous Coronary Intervention

Rupture of an atherosclerotic plaque causes platelet aggregation and thrombus formation common to all types of acute coronary syndromes (ACS) such as unstable angina pectoris, non–ST elevation myocardial infarction (NSTEMI) and ST-elevation myocardial infarction (STEMI), cardiovascular diseases challenging the health care system of many countries. This process is also responsible for abrupt blood vessel closure in patients undergoing percutaneous coronary intervention (PCI). Blood vessel rupture exposes the sub-endothelium to thrombogenic substances and initiates platelet activation and aggregation. Glycoprotein IIb/IIIa receptors (αIIbβ3 integrin) are the most prevalent integrins in the platelets and therefore, their activation is considered the final major pathway for platelet aggregation. Platelet activation promotes conformational changes in the αIIbβ3, permitting fibrinogen to bind with this integrin. Each fibrinogen molecule has two αIIbβ3 binding sites enabling binding to integrin receptors on two neighboring platelets, property contributing to the formation of platelet-rich arterial thrombi. During healthy conditions, the circulating platelets do not bind to blood vessels’ endothelium. Upon vessel injury, exposure of the sub-endothelial ECM proteins promotes adhesion of platelets surrounding the injury. Of particular importance to platelet adhesion are the glycoproteins GP Ib/V/IX, GP VI, and GP Ia/IIa receptors (α2β1 integrin). von Willebrand factor (vWF), which is inactive, is deposited constitutively from endothelial cells as part of the ECM proteins and is also secreted from activated endothelial cells. Following blood vessel injury, vWF is deposited on collagen fibers exposed at the injury location. As the vWF-GP Ib complex is relatively weak, supplementary adhesion mediated by platelet integrins is also provided for strong platelet attachment to the injury location. Platelet adhesion is a process characterized by several steps. First, shear forces rapidly generated as blood circulates over endothelial injury location initiate temporary platelet-endothelium interactions. The key element of this critical step is the weak binding of GP Ib to vWF generating multiple platelet membrane extensions. In the next step, the low-affinity interaction between collagen and GP VI platelets causes platelet activation and aggregation, a process enhanced by α2β1 integrin. The process of platelet activation and aggregation is further enhanced by paracrine and autocrine-mediated signaling through the release of thromboxane A_2_ (TxA2) and adenosine diphosphate (ADP) from platelets, along with thrombin activation by the blood vessel wall’s tissue factor (Figure 2). Further reinforcement of stable platelet adhesion is central to the interaction ofαIIbβ3 integrin with fibrinogen and vWF. Moreover, the hemorrhage at the injury location is initiating the generation of thrombin, which is a potent platelet activator (Figure 2). Thrombin acts to convert fibrinogen to fibrin, which is used as a stable lattice for the developing platelet plug thrombus. This thrombus significantly provides occlusion of the coronary artery in myocardial or brain ischemia. Drugs of “disintegrin” peptide family containing the Arg-Gly-Asp (RGD) amino acid sequence, Eptifibatide, and Tirofiban (Figure 1, Table 1) by blocking αIIbβ3 integrin, are currently considered the most powerful specific inhibitors of platelet aggregation in acute thrombosis. Since the hemostatic function of platelets is also dependent on this integrin, this antiplatelet therapy comes with bleeding risks. 

Clinical trials have demonstrated the efficacy of αIIbβ3 integrin antagonists in patients with acute coronary syndrome (ACS) based on risk stratification (the process of separating patient population groups according to the degree of risk). High-risk patients with unstable angina/NSTEMI myocardial infarction derive particular benefit from αIIbβ3 integrin antagonists and an early invasive strategy [19]. The 2000 ACC/AHA unstable angina and NSTEMI myocardial infarction (MI) guidelines recommend therapy with αIIbβ3 integrin antagonists in patients with NSTEMI acute coronary syndromes (ACS) at high risk of death or a nonfatal MI. Such high-risk conditions include ST-segment depression on ECG, positive high levels of troponin I, indications of left ventricular dysfunction, pulmonary edema, age greater than 75 years old, ongoing chest pain for greater than 20 minutes, or ischemia refractory to other treatments. Additionally, αIIbβ3 integrin antagonists are utilized for patients who undergo a percutaneous coronary intervention (PCI), such as balloon angioplasty, atherectomy, or stent implantation. However, only Eptifibatide (Integrilin^®^) has FDA labeled support for both of these clinical indications. The αIIbβ3 integrin antagonists have similar mechanisms of action to inhibit the platelet aggregation process. Abciximab, the chimeric monoclonal antibody blocks binding of not only αIIbβ3 integrin important for platelet aggregation, but also other integrins such as αVβ3 vitronectin receptor also involved in platelet, and therefore, may result in more bleeding adverse effects. Tirofiban and Eptifibatide are small, synthetic molecules developed from the snake venom RGD-disintegrins, which have high affinity and relative selectivity for the αIIbβ3 integrin. These compounds compete with fibrinogen for the αIIbβ3 integrin receptor in a dose-dependent manner and prevent platelet aggregation. The clinical therapeutic goal of using these drugs is to achieve 80% inhibition of platelet aggregation with minimal bleeding adverse effects.

### 1.5. Tirofiban (Aggrastat)

The first antiplatelet drug derived from a snake venom protein was Tirofiban. Tirofiban is a non-peptide molecule that was developed based on the RGD motif present in the parent disintegrin molecule, Echistatin (Figure 1).

Echistatin was first isolated in 1988 from the venom of the saw-scaled viper and found as an effective antagonist of fibrinogen-induced platelet aggregation [20]. This disintegrin contains an RGD-sequence in an exposed loop which binds to αIIbβ3, αvβ3, αvβ5, and α5β1 integrins with very high affinity in the low nanomolar range [IC_50_-values: αvβ3 (0.46 nM), α5β1 (0.57 nM), and αIIbβ3 (0.9 nM)]. Considering that the distance between the positively charged argininyl and the negatively charged aspartyl centers of RGD is a key determinant of potency, a tyrosine analog was developed by Merck Co. Through optimization of the RGD-motif lead peptide using, at the NH_2_-terminal, a 4-(4-piperidinyl) butyl group and, at the COOH-terminal, an (S)-butylsulfonylamino group, a peptidomimetic was developed with a 3000-fold increase in potency for inhibition of platelet aggregation, with preservation of selectivity for αIIbβ3 over other integrins. This compound, initially designated MK-0383, was very effective in inhibiting αIIbβ3 function in animal models [21] and finally, named Tirofiban, pharmacologically defined as a low–molecular weight (<1 kDa), reversible antagonist for clinical use [22]. Tirofiban does not inhibit αVβ3 or αMβ2 integrins. When administered according to the protocol of 0.4 µg/kg/min for 30 minutes, followed by a 0.1 µg/kg/min maintenance infusion, 90% inhibition of platelet aggregation was obtained by the end of the initial infusion. Its plasma half-life is 2 h, and following discontinuation of an infusion of Tirofiban, 0.1 µg /kg/min, ex vivo platelet aggregation returns to near baseline in 4–8 h in approximately 90% of patients. Tirofiban is eliminated by renal and biliary excretion [23]. The clinical trials that investigated the efficacy of Tirofiban for therapy of patients with ACS (unstable angina/ NSTEMI myocardial infarction) were RESTORE, PRISM, PRISM-PLUS, which differed in their drug regime and number of investigated patients (Table 2).

The one-month reduction in mortality in these clinical trials was 16% relative reduction (*p* = 0.160) in RESTORE and 36% (2.3 versus 3.6%; *p* = 0.02) in PRISM and 27% (8.7 versus 11.9%; *p* = 0.027) in PRISM-PLUS. In the PRISM trial, there was no difference in bleeding times between the Tirofiban and placebo groups, and bleeding increased only modestly in the PRISM-PLUS (1.4 versus 0.8%; *p* = 0.23) Tirofiban together with Heparin compared to Heparin-alone groups and in RESTORE (5.3% versus 3.7%; *p* = 0.096) Tirofiban compared to placebo groups (Table 2) [24,28]. The TARGET trial was primarily designed to test whether Tirofiban was not inferior to Abciximab in patients undergoing PCI. The primary endpoint included death, MI, and target vessel revascularization (TVR) within 30 days after PCI. Overall, 4809 patients were randomized and received the study drug. The incidence of the primary endpoint was 7.6% in the Tirofiban group and 6.0% in the Abciximab group, representing a significant difference of 27%. The result of the test for equivalence was not statistically significant, while the test for superiority of Abciximab did. At 1-year follow-up, the mortality rate did not differ significantly between the two groups [34].

### 1.6. Eptifibatide (Integrilin)

In order to discover a selective disintegrin for αIIbβ3, several dozens of venoms were screened, leading to the discovery of Barbourin (Figure 1), purified from the venom of the snake *Sistrurus miliarius barbouri* [35]. The disintegrin, Barbourin, isolated from this Southeastern pigmy rattlesnake, contains an amino acid substitution of Lys (K) for Arg (R) in the RGD sequence resulting with a KGD motif highly specific for αIIbβ3 (GP IIb-IIIa) [36]. Using this information, a series of conformational constrained, disulfide-bridged peptides were synthesized, containing the KGD amino acid sequence. Incorporation of the KGD sequence into a cyclic peptide template, followed by systematic optimization of the cyclic ring size, hydrophobicity, and the derivatization of the lysine side chain of the KGD sequence yielded peptide analogs which displayed αIIbβ3 integrin inhibitory potency and selectivity, comparable to that of Barbourin [37]. Eptifibatide (Integrilin), one of the derivatives of Barbourin (Figure 1), is a cyclic heptapeptide, competitive antagonist for the activated, platelet αIIbβ3 integrin using the KGD integrin recognition sequence [38]. Its mechanism of action is the prevention of the binding and cross-linking of fibrinogen to the platelet surface, causing inhibition of platelet aggregation and preventing thrombus formation. Through a series of small preclinical and clinical trials, an effective dose regimen was determined. Modeling of the two-compartment drug’s pharmacokinetics established the importance of a double-bolus upon starting the drug treatment. In a large-scale clinical trial using this double-bolus approach, in PCI procedures, the therapeutic efficacy was shown to be significantly improved [39]. To date, in addition to the dual antiplatelet therapy using Aspirin (cyclooxygenases inhibitor) and Clopidogrel (irreversible inhibitor of purinergic P2Y12 receptor) (Figure 2) and systematic stent implantation, the use of the Eptifibatide, proved beneficial in improving the early outcome of PCI, especially in higher-risk clinical and/or anatomical subsets. In healthy volunteers and ACS patients undergoing PCI, the drug potently inhibited ex vivo platelet aggregation as well as fibrinogen binding to adenosine diphosphate (ADP)-activated platelets. In patients with ACS, the onset of ADP-induced platelet aggregation inhibition was 5 minutes after starting Eptifibatide infusion, persisted for the duration of the infusion period and returned to normal values within 4–8 h. The PURSUIT clinical trial (Table 2), conducted in >10,000 patients with unstable angina or NSTEMI myocardial infarction (MI), indicated that the reduction in the end-point of >80% inhibition of platelet aggregation has been achieved with a bolus of 180 μg/kg and using an infusion rate of 2 μg/kg/min. The dosing protocol used in the ESPRIT study (Table 2) was similar to that used in the PURSUIT study (a 180 μg/kg bolus followed by a 2.0 μg/kg/min infusion), but added a second 180 μg/kg bolus ten minutes after the first bolus to prevent decrease in platelet aggregation inhibition, before reaching steady-state using the protocol of continuous 2.0 μg/kg/min infusion. This dosing regime was recommended in order to maintain platelet aggregation inhibition above 80% immediately after the performance of PCI. Eptifibatide increases bleeding times by 2-, 4-fold compared with that in patients with unstable angina or NSTEMI MI as well as those undergoing PCI. Bleeding time returns to baseline 1 h after stopping the infusion. In the clinic, the drug is used in combined therapy as an adjunct to either Heparin (which binds to the enzyme inhibitor antithrombin III, causing inactivation of thrombin, factor Xa and other coagulation proteases) or Aspirin (Figure 2). The drug is effective in patients undergoing PCI, whether or not they have unstable angina or NSTEMI MI. In the IMPACT-II trial, when measured per-protocol (patients treated), but not intent-to-treat, analysis, at a dosage of 135 μg/kg followed by 0.5 μg/kg/min for 24 h, Eptifibatide reduced the monthly risk of the end-point (combined data on death, nonfatal MI and urgent or emergency coronary interventions) by 2.5% (absolute reduction) in patients undergoing PCI. The drug also reduced the incidence of blood vessel closure and ischemic cardiovascular complications in the first day of greatest risk. Bleedings events are the most common adverse effects following Eptifibatide therapy. The drug is not immunogenic and not associated with an excess of brain bleeding, stroke or thrombocytopenia, and did not increase the bleeding risk in patients undergoing coronary artery bypass graft. Eptifibatide delivered intravenously, when combined with Aspirin and Heparin, reduces the 30-day risk of ischemic events in patients with unstable angina and NSTEMI MI and decreases ischemic cardiovascular complications at the time of greatest risk in patients undergoing PCI. Eptifibatide may improve coronary flow when combined with thrombolytic drugs in patients with acute MI, but the possibility of aggravating bleeding by this drug combination should be remembered [40]. With its acceptable acute tolerability, this drug is an option as a short-term adjunct drug in the above mentioned clinical settings. Importantly, dosage adjustment is not required in geriatric patients or in patients with renal dysfunction (serum creatinine <2 mg/dL). Due to drug-drug interactions, caution should be taken when administering Eptifibatide with other drugs. The drug is contraindicated in patients with bleeding abnormalities, brain stroke, severe high blood pressure or severe renal disablement. Increase in drug exposure was linearly related to increases in administered doses and steady state was achieved within 4–6 h. This drug was bound by 25% to plasma proteins and most of it is excreted in the urine and recovered as the parental drug and several metabolites. No major Eptifibatide metabolites have been detected in human blood. About 40% of total body clearance occurred by renal mechanisms. Systemic clearance of the drug was 9.1 L/h in patients undergoing PCI. Eptifibatide clearance was decreased and blood concentrations were increased in patients with severe kidney impairment (serum creatinine >2 to 4 mg/dl) and in aged patients (>60 years). The elimination half-life was 2.5–2.8 h in patients undergoing PCI.

### 1.7. Platelet’s GP Ia/IIa (α2β1) Integrin Receptor for Collagen

Collagen, an ECM compound with different structural characteristics is a group of proteins composed of different types and synthesized by a wide variety of cells. In blood vessels, it represents about 40% of the total protein content and its role is to maintain wall integrity and elasticity. All types of collagens can induce in vitro platelet aggregation. Among the different collagens present in the blood vessel wall, type I, III (fibrillary collagen) and IV (non-fibrillary collagen) are the most reactive on platelet adhesion, activation, and aggregation. Collagen types I and III are largely exposed to platelets aggregation when the blood vessel wall injury extends to the deeper layers of media and adventitia. Collagens are strong inducers of platelets aggregation. GP Ia/IIa and GP VI are the two well-known collagen receptors present on platelets (Figure 2) playing important role in thrombosis and therefore, used as targets for the development of novel drugs to prevent collagen-mediated thrombosis [41]. Glycoprotein Ia/IIa receptor also referred to as integrin α2β1 (VLA-2, or CD 49b/CD 29), binds to collagen, facilitating platelet adhesion and aggregation. Fibrillary collagens are the best ligands for the α2β1integrin receptor. Collagen and vWF cooperate synergistically to reinforce platelet adhesion at the sites of blood vessel wall injury. Various studies confirm the importance of α2β1in platelet activation and thrombosis: i. reduced expression of α2β1 on the platelet is associated with the extension of bleeding times and reduced adhesion to collagen; ii. increase levels of α2β1has been observed in systemic sclerosis patient’s platelets; iii. α2β1modulates platelet adhesion and spreading on collagen; iv. studies on Cre/loxP-mediated loss of β1 integrin on platelets emphasized the importance of α2β1in vascular thrombosis; v. epidemiologic studies found a direct correlation between a high expression of α2β1and elevated risk of MI and stroke; vi. genetic polymorphism in both GP VI and α2β1have been proposed to be associated with predisposition to thrombosis. It is accepted that major role of α2β1is to increase the affinity of collagen for the platelet’s membrane surface and enhance the activation of GP VI, which regulates the primary response to blood vessel injury, while α2β1possibly mediates the secondary adhesive response. [41,42,43,44]. Snake venom proteins such as C-lectin type proteins, which inhibited platelet aggregate dimensions as well as adhesion, have been widely used in investigating collagen-mediated signaling in platelets and towards anti-platelets drug development [45].

### 1.8. Vipegitide, a Partial Antagonist of α2β1 Integrin with Antiplatelet Activity

Despite recent progress in the development of antithrombotic drugs, there is an unmet clinical need for effective new drugs to prevent and treat cardiovascular diseases, while minimally impairing physiological hemostasis. In particular, drugs that target platelets involved in the formation of thrombi would specifically affect atherosclerotic lesions but with a reduced risk of bleeding side effects. Therefore, α2β1 integrin is a good target for novel antithrombotic therapy because it is associated with high cardiovascular risk of stroke and myocardial infarction [46] and since its overexpression is associated with pathological clot formation whereas its absence does not cause severe bleeding. α2β1 is expressed at low levels in a mildly prolonged bleeding time clinical situations, which is quite different from the profound bleeding disorder observed in deficiency of the platelet fibrinogen receptor integrin αIIbβ3 (Glanzmann’s thrombasthenia) [47]. Despite the supportive role that α2β1 integrin seems to play in collagen-mediated platelet adhesion, there is relatively poor drug development targeting platelets α2β1 with respect to arterial thrombosis [48]. Vatelizumab, a humanized α2β1-blocking antibody was recently investigated for its safety and efficacy during a Phase 2 clinical study in multiple sclerosis patients, since α2β1 integrin-mediated collagen binding at the site of inflammation is cardinal to a number of pro-inflammatory processes in hematopoietic cells, including platelets [49]. Some small molecules that inhibit integrin α2β1 have been found, including Saratin [50], and lipophilic molecules that prevent pathological thrombus formation [51,52,53]. However, to the best of our knowledge, we are heedless of any clinical trial investigating small drug molecules targeting inhibition of the platelets α2β1 integrin. Important snake venom protein family known to affect platelet function is the C-type lectin-like protein (CTL). Different CTLs exhibit a number of functional activities related to hemostasis, including binding to Factor IX and X to inhibit blood coagulation, inhibit the binding of thrombin to fibrinogen, and inhibit or activate platelet aggregation by interacting with vWF or collagen receptors. The CTLs basic structure consists of the subunits α and β, which form heterodimers via a typical domain-swapping motif. Rhodocetin is a selective α2β1 integrin antagonist consisting of four distinct subunits [54]. Conjointly with EMS16 [55] and VP12 [56,57], Rhodocetin inhibits collagen-binding to the α2A-domain. These α2β1 integrin-specific antagonists are highly selective, natural lead compounds for the development of anti-platelets, anti-metastatic and anti-angiogenic drugs [58]. We would like to emphasize this approach by the anti-platelet lead compound Vipegitide (Figure 1) [59] developed in our preclinical studies using C-type lectins from the venom of *Vipera palestinae* [56,57,58,59]. The venom was separated by high-pressure liquid chromatography (HPLC). Screening of the fractions, using cell adhesion assays of cells overexpressing α2β1 integrin, identified a selective antagonist of α2β1 integrin which was purified to a protein named VP 12. The amino acid sequence of the two subunits of VP 12 was established by mass spectrometry and N-terminal sequencing of their proteolytic fragments. For this purpose, the VP 12 molecule was reduced and the ethylpyridylated (EV) subunits were separated by HPLC and named VP 12A and VP 12B. Thereafter, MALDI-TOF mass spectroscopic analysis of unmodified VP 12 yielded a single molecular ion of 30,387 Da, whereas EP-VP 12A and EP-VP 12B subunits showed 15,981 Da and 15,893 Da, respectively. Based on the structure of other C-lectin type protein isoforms, which have similar activity to VP 12, we predicted that each subunit contained an equal number of seven cysteines. Complete amino acid sequences of both subunits of VP 12 were established using a standard procedure including N-terminal sequencing of separated EP-subunits and their tryptic, overlapping fragments, confirming that VP 12 belongs to the C-type lectin-related protein family. In another approach, to specifically isolate additional antagonists targeting the α2β1 integrin, we performed a protocol based on affinity chromatography using the extracellular recombinant α2β1 integrin-A domain, immobilized to a resin, and isolated a novel CTL protein named VP-i [57]. We found that VP-i binds to the α2 integrin A domain and that it significantly inhibited adhesion of various cells to type I collagen and inhibited cell migration. Moreover, we found that VP-i differed structurally from the previously purified VP 12, although not functionally, and therefore represented another venom lead compound that can be utilized for further drug development [57]. In the next steps, linear and folded peptides containing the integrin binding motif W^1^KTSRTSHY^9^ were used as a template to prepare a novel peptidomimetic antagonist of α2β1 integrin, with platelet aggregation-inhibiting activity, named Vipegitide [59]. Vipegitide is a 13-amino acid, conformation restricted peptidomimetic molecule, containing two α-aminoisobutyric acid residues. Since it was not stable in human serum, by substitution of glycine and tryptophan residues at positions 1 and 2 with a unit of two polyethylene glycol (PEG) we generated a novel analog, Vipegitide-PEG2, which was stable in human serum for over 3 h. Vipegitide and Vipegitide-PEG2 were characterized by high potency (7 × 10^10^ M and 1.5 × 10^10^ M, respectively) and intermediate efficacy (40% and 35%, respectively) as well as selectivity toward α2 integrin. Binding of these peptidomimetics with a recombinant extracellular α2 integrin A domain was confirmed in a cell-free receptor binding assay. Vipegitide and Vipegitide-PEG2 inhibited α2β1 integrin-mediated adhesion and aggregation of human and murine platelets under static and flow conditions by 50%. They potently inhibited collagen I-induced platelet aggregation in platelet-rich plasma and whole human blood. Higher potency of Vipegitide compared to Vipegitide-PEG2 was correspondent with results of computer modeling in water. These peptidomimetics were acutely tolerated in mice upon intravenous bolus injection of 50 mg/kg [59]. These findings underline the further use of Vipegitide and Vipegitide-PEG2 as platelet aggregation-inhibiting tools in antithrombotic therapy.

## 2. Conclusions

The research and development of integrin antagonists is a good example of a translational medicine approach whereby the discovery of the naturally occurring disintegrins and C-type lectin proteins in snake venoms has inspired much research into the molecular interaction of RGD/KGD/RTS/KTS disintegrins with integrin αIIbβ3 andα2β1, leading to drug development of small-molecule that were successful in clinical trials. In turn, the results and adverse effects observed in these clinical trials have enriched our understanding of the pathophysiology of different cardiovascular diseases. Antiplatelet therapy (Figure 2) is a cornerstone in the prophylaxis of myocardial infarction and stroke. Currently, blockade of the most important and highly expressed platelet receptor, the fibrinogen-binding αIIbβ3-integrin is an integral drug of acute coronary syndrome therapy. Additionally, the intravenous administered Eptifibatide and Tirofiban αIIbβ3 antagonistic drugs are used as an adjunct therapy in high-risk coronary interventions. They efficiently block fibrinogen-binding preventing the most crucial mechanical step of aggregation and therefore characterized as the strongest antithrombotic drugs. It is interesting to note that, in contrast to other common anti-platelet drugs, such as Aspirin and Clopidogrel (Figure 2), resistance against or ineffectiveness’s of αIIbβ3 antagonistic has not been described so far. In view of the compelling evidence suggesting a primary role of collagen in the initiation of platelet adhesion and subsequently in intravascular thrombosis, it is therefore important to develop novel collagen—α2β1 integrin targeted antagonists. Despite the advent of novel agents and major advances in antiplatelet treatment over the last decade, αIIbβ3 and α2β1 are still promising therapeutic targets and novel knowhow in the snake venom disintegrin/CTL field should help and accelerate the discovery of novel anti-platelet drugs. 

## Figures and Tables

**Figure 1 toxins-11-00303-f001:**
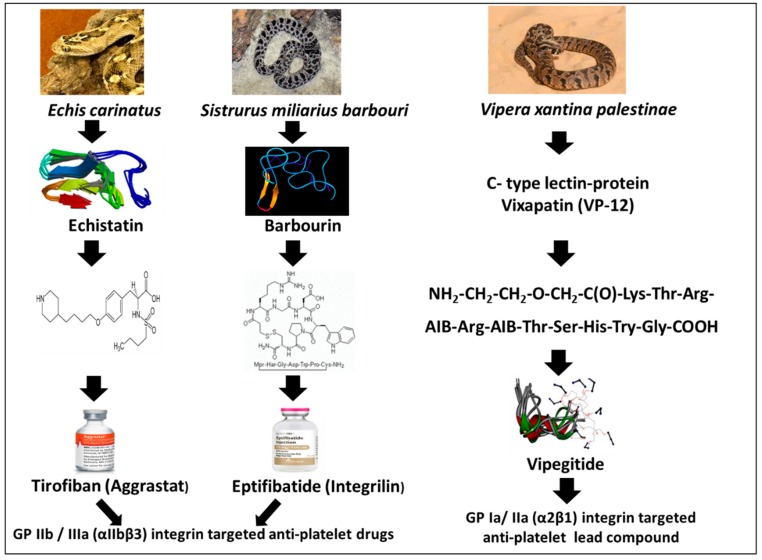
Scheme of anti-platelet drugs and a lead compound developed from snake venom’s disintegrins and C-type lectins.

**Figure 2 toxins-11-00303-f002:**
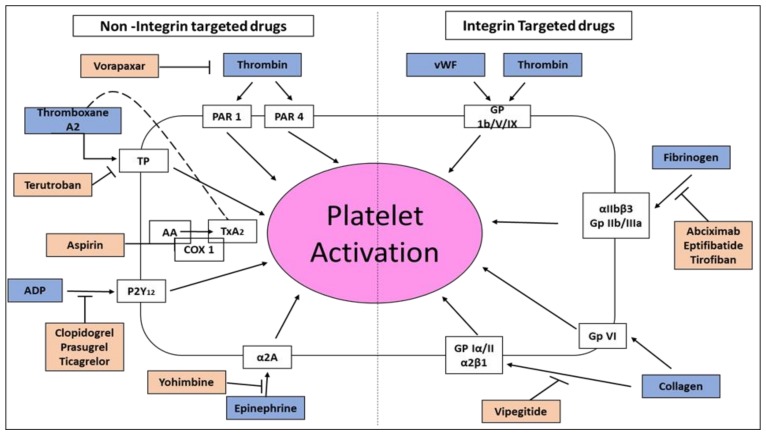
Scheme of antiplatelet therapies using non-integrin and integrin-targeted drugs. Thrombin, thromboxane A2 (TxA2) generated from arachidonic acid (AA) by cyclooxygenase type 1 (COX 1), adenosine diphosphate (ADP), epinephrine (blue boxes, left side ) or biomechanical shear stress forces, by activating their respective receptors (uncolored boxes), result in platelet activation by inside out signaling of integrin αIIbβ3 in cooperation with collagen, fibrinogen, von Willebrand factor (vWF) stimulation of α2β1 integrin, GP VI, αIIbβ3 and GP Ib/V/IX respectively, (blue boxes, right side). Inhibition of TxA2 formation or antagonism of ADP receptor P2Y_12_, thromboxane prostanoid (TP) receptor, thrombin receptor PAR 1,4 or α2_A_ adrenergic receptor (α2A) interrupts positive feedback signaling necessary for sustained platelet activation. Direct inhibition of the integrins αIIbβ3 or α2β1 strongly inhibits platelet activation. Drugs, to name a few, are depicted in orange boxes with inhibitory arrows. Vipegitide is the only compound in preclinical research.

**Table 1 toxins-11-00303-t001:** αIIbβ3 integrin receptor antagonist drugs.

Drug	Product Availability	FDA-Approved Indications ^4^	Clinical Trial
Abciximab ^1^ (ReoPro)	5 mL vial (2 mg/mL)	PCI only	EPIC
Eptifibatide ^2^ (Integrilin)	a. 100 mL (750 µg/mL)	Unstable Angina/NSTEMI/PCI	PURSUIT
	b. 10 mL/vial (2000 µg/mL)		ESPRIT
	c. 100 mL/vial (2000 µg/mL)		IMPACT-II
Tirofiban ^3^ (Aggrastat)	a. 250 mL (50 µg/mL)	Unstable Angina / NSTEMI only	PRISM
	b. 500 mL (50 µg/mL)		PRISM-PLUS
	c. 25 mL/vial (250 µg/mL)		RESTORE
	d. 50 mL /vial (250 µg/mL)		TARGET

^1^ Manufactured by Centocor/Eli Lily, ^2^ Manufactured by COR/Key, Glaxo, Bayer, etc. ^3^ Manufactured by Merck, Abbott, etc.-alternative names: L-462; MK 383; ^4^ PCI = Percutaneous Coronary Intervention (stent, balloon angioplasty, therectomy, rotablation); NSTEMI—non St segment elevation (on the electrocardiography) myocardial infarction.

**Table 2 toxins-11-00303-t002:** Clinical trials evaluating the efficacy and adverse effects of anti-platelet drugs targeting αIIbβ3 integrin in ACS and PCI cardiovascular diseases.

Drug	Description	Indication	Major Clinical Trial Findings *	Major Side Effects	References
Abciximab	Mab targeting αIIbβ3 and inhibiting fibrinogen binding	Prevention of cardiac ischemic effects in patients with ACS undergoing PCI	EPIC (2099 patients): 35% fewer events **	Major bleeding Thrombocytopenia	[24]
			EPILOG (2972 patients): 65% fewer events, benefit 1 year		[25]
			CAPTURE (1265 patients): 29% fewer events		[26]
			EPISTENT (2399 patients): 51% fewer events		[27]
Eptifibatide	A cyclic peptide that blocks the binding of fibrinogen to αIIbβ3 integrin	Patients with NSTEMI who are undergoing PCI	PURSUIT (10,948 patients): 10% fewer events, benefit 6 months	Major bleeding unchanged, minor bleeding slightly increased with Eptifibatide treatment in IMPACT‑II;	[28,29]
			IMPACT II (4010 patients): 19% fewer events		[30]
			ESPRIT (2064 patients): significantly fewer events, benefit 6 months and 1 year		[31]
Tirofiban	A small molecule that that blocks the binding of fibrinogen to αIIbβ3 integrin	Patients with unstable angina or NSTEMI who are undergoing PCI	PRISM (3232 patients): 32% fewer events	Bleeding similar in both groups in PRISM, PRISM-PLUS and RESTORE; Reversible thrombocytopenia was three times more common in Tirofiban-treated patients than placebo-treated patients in PRISM, no difference was observed between these groups in RESTORE	[28]
			PRISM-PLUS (1915 patients): 28% fewer events compared to heparin and aspirin)		[28,32]
			RESTORE (2139 patients): reduction of events at 2- and 7-days post-treatment		[33]

ACS, acute coronary syndrome, including unstable angina; ST-segment elevation myocardial infarction (STEMI) and non-STEMI (NSTEMI); MIDAS, metal ion-dependent adhesion site; PCI, percutaneous coronary intervention; RGD, Arg-Gly-Asp; Mab, monoclonal antibody * Findings are presented as a change in patients treated with drug compared to those treated with placebo, unless otherwise indicated. ** Events describe cardiovascular end-point events, including death, myocardial infarction, repeat PCI, stent or bypass at 30 days.

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
