# Peer review of "From Snake Venom’s Disintegrins and C-Type Lectins to Anti-Platelet Drugs"

_toxins, 2019, doi:10.3390/toxins11050303_

Round 1
Reviewer 1 Report
This review is well written with few English mistakes.
It has a great importance and an update for toxins-based drugs that can lead reads to improve the knowledge to discover new drugs.
Congratulations fo this bright and clear theme review.
Author Response
The authors appreciate the knowledgeble understanding of the reviewer!
Reviewer 2 Report
In this review article the authors summarize several aspects related to drug discovery of anti-platelet agents of snake venom’s origin emphasizing their indications for clinical use in acute coronary syndromes and percutaneous coronary intervention. The information were based on several clinical trials as well as their adverse effects.
The article is well written and give an important overview of this interesting research subject. It describes the fundamental chemical, biochemical and pharmacological and therapeutic properties of so far discovered compounds of these sources. Thus, in my opinion the work deserves publication. However, some aspects must be clarified before its final consideration as a publishable article:
1. As the manuscript describes the interaction between different molecular e cellular components which are involved in the mechanisms of fine regulation of platelets aggregation, it seems that a figure illustrating such relationships would enrichen the article;
2. There is an exaggerated number of keywords;
3. In general, cited references are a little bit old. Is there any reason for that? Has the subject being less explored in most recent years?
4. Due to a large number of abbreviated words, an Abbreviation session seems to be appropriate;
5. In the abstract (lines 5 and 9) as well as in the text (lines 231/232) the expression “resulting with” should be better written as “resulting in”;
6. Lines 182-184: The mention to Figure 1 should be better explained. The sentence does not emphasize, in advance, what the figure intends to show. It seems that the mention to the figure could be as follow for instance: “Figure 1 illustrates the source of the snake venom’s proteins that originates different therapeutic compounds acting as antiplatelet aggregation and antithrombotic drugs”;
7. Several major clinical trial studies mentioned in Table 2 (EPIC, EPILOG, CAPTURE, EPISTENT, ESPIRIT) were not listed in the references;
8. Lines 286-302: the text should be rewritten in order to reduce the number of sentences starting with the word Eptifibatide;
9. Lines 320-327: Which references support these observations?
10. Lines 334-406: Most of the text in these lines is related to the characterization of Vipegitide and its physio-pharmacological effects. Is this subject better described in already published articles? If so, these references should be included at the beginning of the sentence;
11. Line 462: Reference 13 is numbered twice.
Author Response
1. As the manuscript describes the interaction between different molecular e cellular components which are involved in the mechanisms of fine regulation of platelets aggregation, it seems that a figure illustrating such relationships would enrichen the article;
Reply: We appreciate your sugestion and added Figure 2 to revised manuscript to illustrate these mechanisms
2. There is an exaggerated number of keywords;
Reply: We corrected the key words
3. In general, cited references are a little bit old. Is there any reason for that? Has the subject being less explored in most recent years?
Reply: Indeed some of the references relevant to drug development of Integrilin and Tirofiban are old but authentic and represent the key studies in the development of these drugs.
4.Due to a large number of abbreviated words,an Abbreviation session seems to be appropriate;
Reply: An abbreviation paragraph was added in corrected draft.
5. In the abstract (lines 5 and 9) as well as in the text (lines 231/232) the expression “resulting with” should be better written as “resulting in”;
Reply:Due to changes requested by similarity report this change is not relevant!
6. Lines 182-184: The mention to Figure 1 should be better explained. The sentence does not emphasize, in advance, what the figure intends to show. It seems that the mention to the figure could be as follow for instance: “Figure 1 illustrates the source of the snake venom’s proteins that originates different therapeutic compounds acting as antiplatelet aggregation and antithrombotic drugs”;
Reply: Thank you!your explanation was inseted in the text!
7. Several major clinical trial studies mentioned in Table 2 (EPIC, EPILOG, CAPTURE, EPISTENT, ESPIRIT) were not listed in the references;
Reply:Thank you! All clinical trials references are now presented in the foot notes of Table 2!
8. Lines 286-302: the text should be rewritten in order to reduce the number of sentences starting with the word Eptifibatide;
Reply: We reduced the number of Eptifibatide enumerations.
9. Lines 320-327: Which references support these observations?
Reply: Reference 33
10. Lines 334-406: Most of the text in these lines is related to the characterization of Vipegitide and its physio-pharmacological effects. Is this subject better described in already published articles? If so, these references should be included at the beginning of the sentence;
Reply:Done.
11. Line 462: Reference 13 is numbered twice.
Reply: The mistake was corrected.